# Development of a denoising convolutional neural network-based algorithm for metal artifact reduction in digital tomosynthesis for arthroplasty: A phantom study

Tsutomu Gomi[ORCID]*, Rina Sakai, Hidetake Hara, Yusuke Watanabe, Shinya Mizukami

School of Allied Health Sciences, Kitasato University, Sagamihara, Kanagawa, Japan

* gomi@kitasato-u.ac.jp

**Data Availability Statement:** All relevant data are within the manuscript.

**Funding:** The author(s) received no specific funding for this work.

## Abstract

The present study aimed to develop a denoising convolutional neural network metal artifact reduction hybrid reconstruction (DnCNN-MARHR) algorithm for decreasing metal objects in digital tomosynthesis (DT) for arthroplasty by using projection data. For metal artifact reduction (MAR), we implemented a DnCNN-MARHR algorithm based on a training network (mini-batch stochastic gradient descent algorithm with momentum) to estimate the residual reference (140 keV virtual monochromatic [VM]) and object (70 kV with metal artifacts) images. For this, we used projection data and subtracted the estimated residual images from the object images, involving hybrid and subjectively reconstructed image usage (back projection and maximum likelihood expectation maximization [MLEM]). The DnCNN-MARHR algorithm was compared with the dual-energy material decomposition reconstruction algorithm (DEMDRA), VM, MLEM, established and commonly used filtered back projection (FBP), and a simultaneous algebraic reconstruction technique-total variation (SART-TV) with MAR processing. MAR was compared using artifact index (AI) and texture analysis. Artifact spread functions (ASFs) for images that were out-of-plane and in-focus were evaluated using a prosthesis phantom. The overall performance of the DnCNN-MARHR algorithm was adequate with regard to the ASF, and the derived images showed better results, without being influenced by the metal type (AI was almost equal to the best value for the DEMDRA). In the ASF analysis, the DnCNN-MARHR algorithm generated better MAR compared with that obtained employing usual algorithms for reconstruction using MAR processing. In addition, comparison of the difference (mean square error) between DnCNN-MARHR and the conventional algorithm resulted in the smallest VM. The DnCNN-MARHR algorithm showed the best performance with regard to image homogeneity in the texture analysis. The proposed algorithm is particularly useful for reducing artifacts in the longitudinal direction, and it is not affected by tissue misclassification.

**Competing interests:** The authors have declared that no competing interests exist.

## Introduction

Cementless hip arthroplasty has gained more popularity in clinic, recently. It is essential that biological fixation procedures employed are reliable for achieving success with this technique [1]. Medical imaging plays an important role for assessing the proper placement of the components of hip arthroplasty, postoperatively, and also to evaluate the potential complications in the long-term [2]. Digital tomosynthesis (DT), a recently developed technique, provides three-dimensional (3D) structural information to a limited extent, by combining computed tomography (CT) with the advantages of digital imaging [1–8], and another advantage of DT is that it can be employed easily with radiography, it can help reducing the radiation doses. However, the image rebuilding procedure using DT is unpredictable and is restricted by low ratios of signal-to-noise, related to the superposition of multiple low-exposure projection images.

Metal objects, that reduce the image quality by decreasing contrast and masking specific features, obstruct the observation of relevant organ parts leading to incorrect diagnosis. Before imaging, it is necessary to ascertain that there is no hematoma or inflammation in tissues surrounding target area and to evaluate any potential interaction of osteosynthetic materials, metallic joint prostheses or implants with nearby tissues and radiation.

Artifacts in DT imaging show along the sweep direction, as zones of much less signal, surrounding the edges of metal prostheses and osteosynthetic materials, which are highly attenuating. This is mostly due to discrepancies between the reality (i.e., wide spectral range) and the reconstruction algorithm assumptions (i.e., ideal monochromatic beam). A relatively minor contribution to these artifacts is also caused by limited sweep angle.

Efficiency of iterative reconstruction (IR) was investigated in earlier studies on DT for arthroplasty [1, 6, 7]. Indeed, the image quality was superior and the balance between low- and high-frequency features was better with IR, compared with the filtered back projection (FBP) [5] technique [1, 7]. In fact, many of the earlier studies made a quantitative comparison of radiation doses generated by different prevailing DT algorithms for arthroplasty and the image qualities [7, 9] and have noticed that IR effectively lowers both radiation exposure and quantum noise. In the latest study [9], it was found that among IR algorithms (including total variation [TV]-based compressive sensing [10–13]), the reconstruction algorithm with the best effect for reducing metal artifacts in DT imaging was maximum likelihood expectation maximization (MLEM) [14].

Previous reports have evaluated metal artifacts and have developed methods (adaptive filtering [combined IR and shift-and-add method] using polychromatic X-ray [15] and combined material decomposition and adaptive filtering using dual-energy [DE] X-ray [16]) for metal artifact reduction (MAR) [15–18]. Among these reported MAR methods, the most effective method for reducing metal artifacts at present is the DE material decomposition reconstruction algorithm (DEMDRA) [16]. Although the DEMDRA is particularly excellent for reducing the high-frequency component in a metal artifact image, its drawback involves mechanical limitations (DE X-ray exposure) because it requires material decomposition processing using DE X-rays. Therefore, further studies are required to generalize the benefits of MAR.

Deep learning approaches have successfully been employed recently, in pattern recognition and image processing methods, including image denoising [19], image super-resolution [20], and low-dose CT reconstruction [21, 22]. For instance, a convolutional neural network (CNN) has been implemented for artifact reduction in medical imaging [23, 24], and the CNN has been employed to get rid-off the residual errors from MAR. Even though these previous studies showed that the CNN could enhance MAR effectively, no study has been conducted on MAR using DT. A CNN-based modification (denoising convolutional neural network

[DnCNN]) was presented by Zhang et al. [19]. The feature of this DnCNN is construction to include the progresses in learning algorithms, very deep architecture and methods of regularization for image denoising. The reference image used in the training workflow for the DnCNN is important for enhancing MAR. Different physical elements can generate metal artifacts, and these include beam hardening, photon starvation, and X-ray scattering. Beam hardening results when an X-ray beam consisting of polychromatic photons passes through a medium. It was suggested earlier that DE virtual monochromatic (VM) spectral imaging can potentially reduce beam hardening induced metal artifacts [16, 25–29]. We think that the application of the VM approach is useful for MAR, as the reference image is appropriate. By performing denoising at the projection data level using DnCNN processing, a reduction effect for metal artifacts after reconstruction can be expected. To support this basis, DnCNN is designed primarily to remove noise from the image. It comprises of a built-in deep feedforward CNN. However, as the DnCNN uses a residual learning method, it is also possible to train the DnCNN architecture to reduce artifacts. In the residual learning method, effects as the MAR method can be expected because the residual image is estimated by learning in the DnCNN network.

The DnCNN algorithm could possibly provide a superior solution to the intrinsic problems. In addition, a decrease in metal artifacts can be achieved by reconstruction employing denoised projection results from each material (e.g., titanium and bone) and adaptive filtering [15]. The novelty of this study is to reduce metal artifacts by processing DnCNN in combination with adaptive filtering [15] at the projection data level. In the present study, we developed a hybrid method of reconstruction that is based on projection space approach by combining the DnCNN and adaptive filtering [15] with a focus on reducing metal artifacts (DnCNN MAR hybrid reconstruction [DnCNN-MARHR] algorithm) in DT. The developmental process of the method and its basic evaluation are presented in this study.

## Materials and methods

### Phantom specifications

A prosthetic phantom consisting of an artificial bone and implant (Table 1) was immersed in the center of a water-filled polymethyl methacrylate case (case dimensions, φ 200 × 300 mm),

**Table 1. Specifications of prosthetic phantom employed in this study.**

|  | Element | Ratio (%) |
|---|---|---|
| **[Artificial bone[a] (Foam cortical shell) ]** | Hydrogen (H) | 7.9192 |
| **Local density (0.48 g/cm³)** | Carbon (C) | 40.4437 |
|  | Nitrogen (N) | 15.7213 |
|  | Oxygen (O) | 35.9157 |
| **[Implant[b] (Titanium alloy) ]** | Titanium (Ti) | 90.255 |
| **Local density (4.43 g/cm³)** | Nitrogen (N) | 0.05 |
|  | Carbon (C) | 0.08 |
|  | Hydrogen (H) | 0.015 |
|  | Iron (Fe) | 0.40 |
|  | Oxygen (O) | 0.20 |
|  | Aluminum (Al) | 5.5 |
|  | Vanadium (V) | 3.5 |

Specifications and chemical composition of the prosthetic phantom employed in this study.

[a]Orthopedic Humerus Normal Anatomy (Model 1013, Sawbones, Inc., WA, USA)

[b]Proximal Retrograde Humeral Nail® (PRHN, Mizuho Inc., Tokyo, Japan)

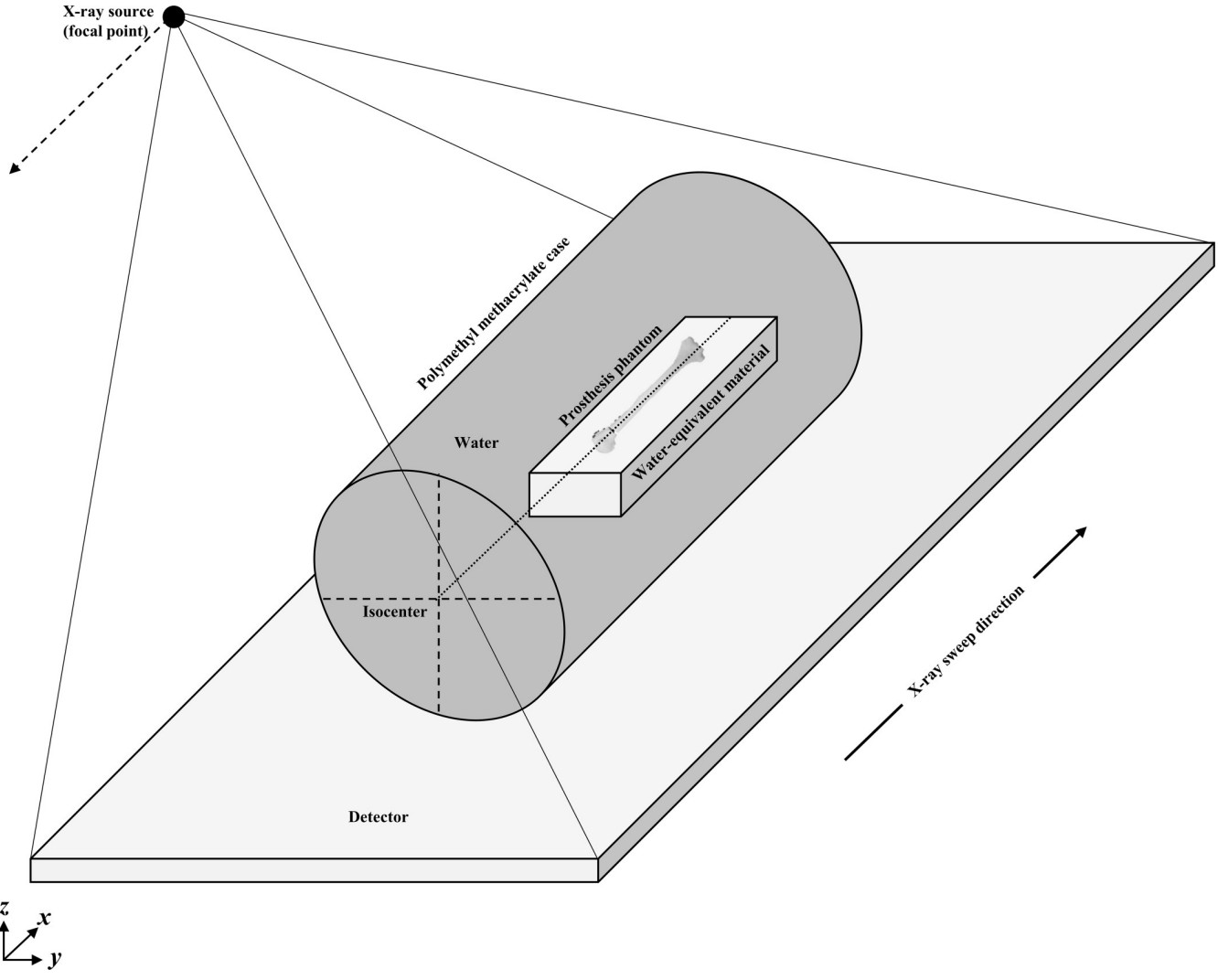

**Fig 1. The experimental geometric placement adopted to assess metal artifact reduction employing the new denoising convolutional neural network metal artifact reduction hybrid reconstruction (DnCNN-MARHR) algorithm and conventional algorithms.** A prosthetic phantom consisting of an artificial joint and implant positioned parallel to the detector plane was employed for the experiments.

for evaluating MAR. A simulated humeral proximal fracture (internal fracture fixation via retrograde intramedullary nail fixation) was present in the prosthetic phantom. In order for DE-DT acquisition, the phantom was positioned parallel to the detector plane (Fig 1).

## DE-DT system

The DE-DT system (SonialVision Safire II; Shimadzu Co., Kyoto, Japan) contained an X-ray tube (anode, made out of tungsten with rhenium and molybdenum; real filter: inherent; aluminum [1.1 mm], additional; aluminum [0.9 mm] and copper [0.1 mm]) with a 0.4-mm focal spot and an amorphous selenium (362.88 × 362.88-mm) digital flat-panel detector (detector element, 0.15 × 0.15-mm). The distances between source (focal point)-to-isocenter and source-to-detector were 924 and 1100 mm, respectively (anti-scatter grid, focused type; grid ratio, 12:1). We selected the kV values (low, 70 kV; high, 140 kV) because the focus of this study is MAR improvement [16].

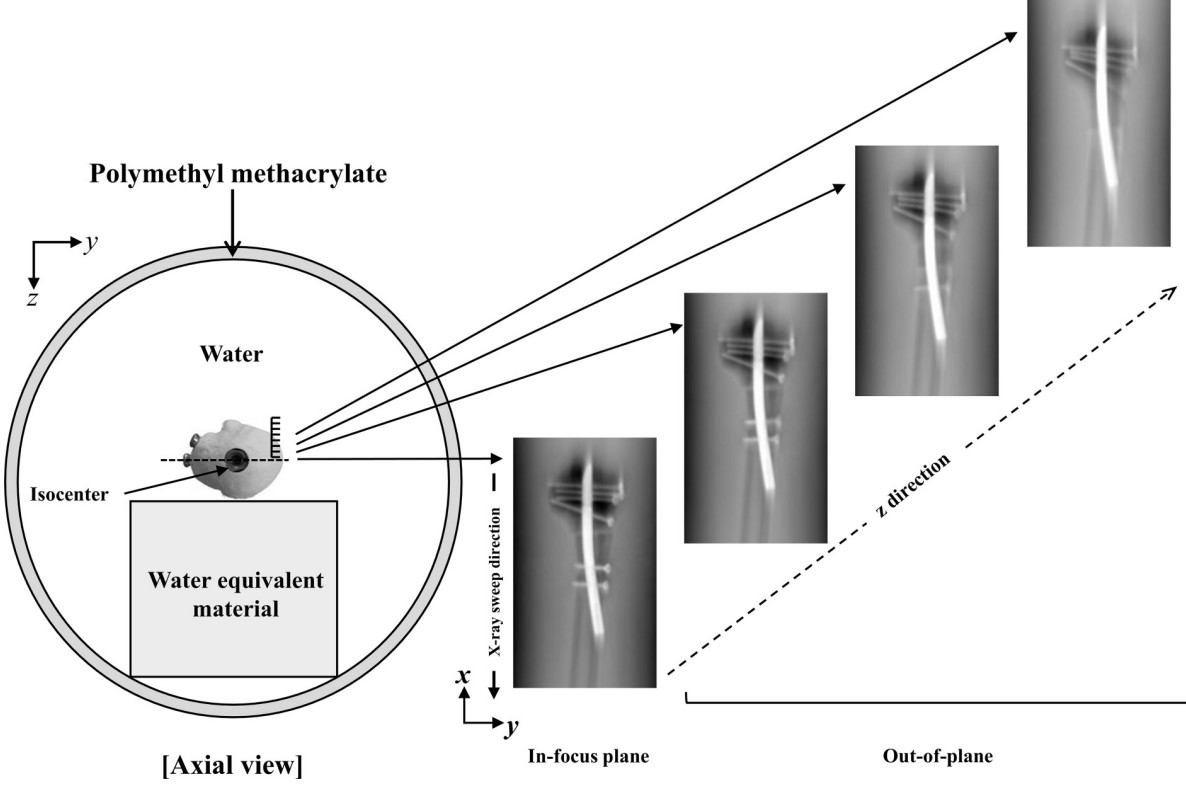

**Fig 2. The schematic diagram to illustrate the relationship of the in-focus plane and the out-of-plane in the Z-axis direction.** The in-focus plane is not affected by the blur, but the out-of-plane is contains blur.

A 40˚ swing angle and linear system movement were employed while conducting tomography, and 37 projection images (1024 × 1024 matrix) at low- and high-voltage were obtained during a single tomographic pass. In DE-DT imaging, pulsed X-ray exposures were used with rapid switching between low and high energies. Even though, low voltage is generally used for clinical application (e.g., prosthesis assessment), all the projection images for low-voltage X-ray were acquired at 187 mA with a 22-ms exposure time and for high-voltage X-ray, at 260 mA with a 5-ms exposure time. To produce reconstructed tomograms of the required height, we employed a 1024 × 1024 matrix with 32 bits (single-precision floating number) per image (pixel size, 0.279 mm/pixel; reconstruction interval, 1 mm; total slice number, 50; starting height of the reconstruction from the detector surface, 150mm). The plane locations were the in-focus plane and out-of-plane from the object location. The image reconstructed at the in-focus plane shows that the object is faithfully reconstructed on the focal plane (Fig 2).

## Generation of reference projection images

One of the approaches for MAR is the use of VM X-ray. A previous study on DT reported that a VM X-ray image is more useful for MAR when compared with a polychromatic X-ray image [16]. Considering these results, we decided to use a VM X-ray image as a reference image for training workflow. The proposed algorithm was developed to realize the projection data level as well as the contents reported by Gomi et al. [16]. In the MAR, it has been reported that DEMDRA applying material decomposition is the most effective for MAR [16]. However, DEMDRA has to apply DnCNN to projection data separated into a plurality of material decompositions. Then, it may be difficult to maintain the residual accuracy with the

polychromatic projection data to be compared. Conversely, VM X-ray imaging can learn residuals with high accuracy by using a single projection datum; thus an effective MAR can be expected. Therefore, the VM X-ray image was used as a reference image in this study. Thirty-seven reference images (VM X-ray projection image) corresponding to the corrected input image pairs were randomly selected as the training set from the original projection data set (total original projection data set: 74). In the present study, we made use of a simple projection space (pre-reconstruction) decomposition method to assess the material fractions ($F_n$) of the artificial bone (foam cortical shell, $F_f$), soft tissue (water, $F_w$), and implant (titanium alloy, $F_t$) in the phantom.

The three basis materials can also be denoted as a linear combination of their attenuation coefficients as follows:

$$\mu(r, E) = \left(\frac{\mu}{\rho}\right)_t (E) \cdot \rho_t(r) + \left(\frac{\mu}{\rho}\right)_w (E) \cdot \rho_w(r) + \left(\frac{\mu}{\rho}\right)_f (E) \cdot \rho_f(r) \tag{1}$$

where the basis materials exhibit different photoelectric and Compton characteristics; $(\mu/\rho)_i$ $(E)$, $i = t$ (titanium alloy), $w$ (water), and $f$ (foam cortical shell), is the mass attenuation coefficient of the three basis materials; and $\rho_i(r)$, $i = t$ (titanium alloy), $w$(water), and $f$(foam cortical shell), is the local density ($g/cm^3$) of the three basis materials at location $r$.

In DE acquisition, the detected image intensity can be depicted as:

$$P_L = \int I_L(E) \exp\left\{ -\left(\frac{\mu}{\rho}\right)_t (E) \cdot K_t - \left(\frac{\mu}{\rho}\right)_w (E) \cdot K_w - \left(\frac{\mu}{\rho}\right)_f (E) \cdot K_f \right\} dE \tag{2}$$

$$P_H = \int I_H(E) \exp\left\{ -\left(\frac{\mu}{\rho}\right)_t (E) \cdot K_t - \left(\frac{\mu}{\rho}\right)_w (E) \cdot K_w - \left(\frac{\mu}{\rho}\right)_f (E) \cdot K_f \right\} dE \tag{3}$$

$$K_t * K_w * K_f = 1.0 \tag{4}$$

where $I_L(E)$ and $I_H(E)$ are the primary intensities at low- and high energy, respectively, whereas, $P_L$ and $P_H$ are the attenuated intensities at low- and high energy, respectively. Each X-ray spectrum is shown in Fig 3. (Measurement tool: RAMTEC413 Toyo Medic Co., Tokyo, Japan; Detector: CdTe; Channel: 1024 (0.2keV/channel); Measuring method: Compton-scattering measurement [30])

The equivalent densities ($g/cm^3$) $K_t$, $K_w$, and $K_f$ of the three basis materials should be calculated for each ray path. Eqs (2), (3) and (4) can be solved for the equivalent area densities, where $K_t$, $K_w$, and $K_f$ represent the unknown materials. Basis material decomposition can thus be ascertained by solving simultaneous equations to determine the values of $K_t$, $K_w$, and $K_f$ from the quantified projection pixel values [31]. By employing the density that corresponds to each of the areas with the 3 basis materials, the linear attenuation coefficient $\mu(r,E)$ can be determined for any photon. The theoretical mass attenuation coefficient and linear attenuation coefficient curve shown in Fig 4 were calculated using the local density and area density of each material. These are generated by inputting the chemical compositions of the titanium alloy, foam cortical shell, and water shown in Table 1 into the XCOM program developed by Berger and Hubbell [32]. Finally, for the projection space decomposition approach, the following process was used to generate material decomposition images for titanium alloy, foam cortical shell, and water.

Eqs (2) and (3) were used to calculate values for $P_{L\_t}$, $P_{L\_w}$, $P_{L\_f}$, $P_{H\_t}$, $P_{H\_w}$, and $P_{H\_f}$ as simulated attenuation intensities of these materials at the two energy levels. These values were then used to construct a sensitivity matrix, and the material fractions (material decomposition

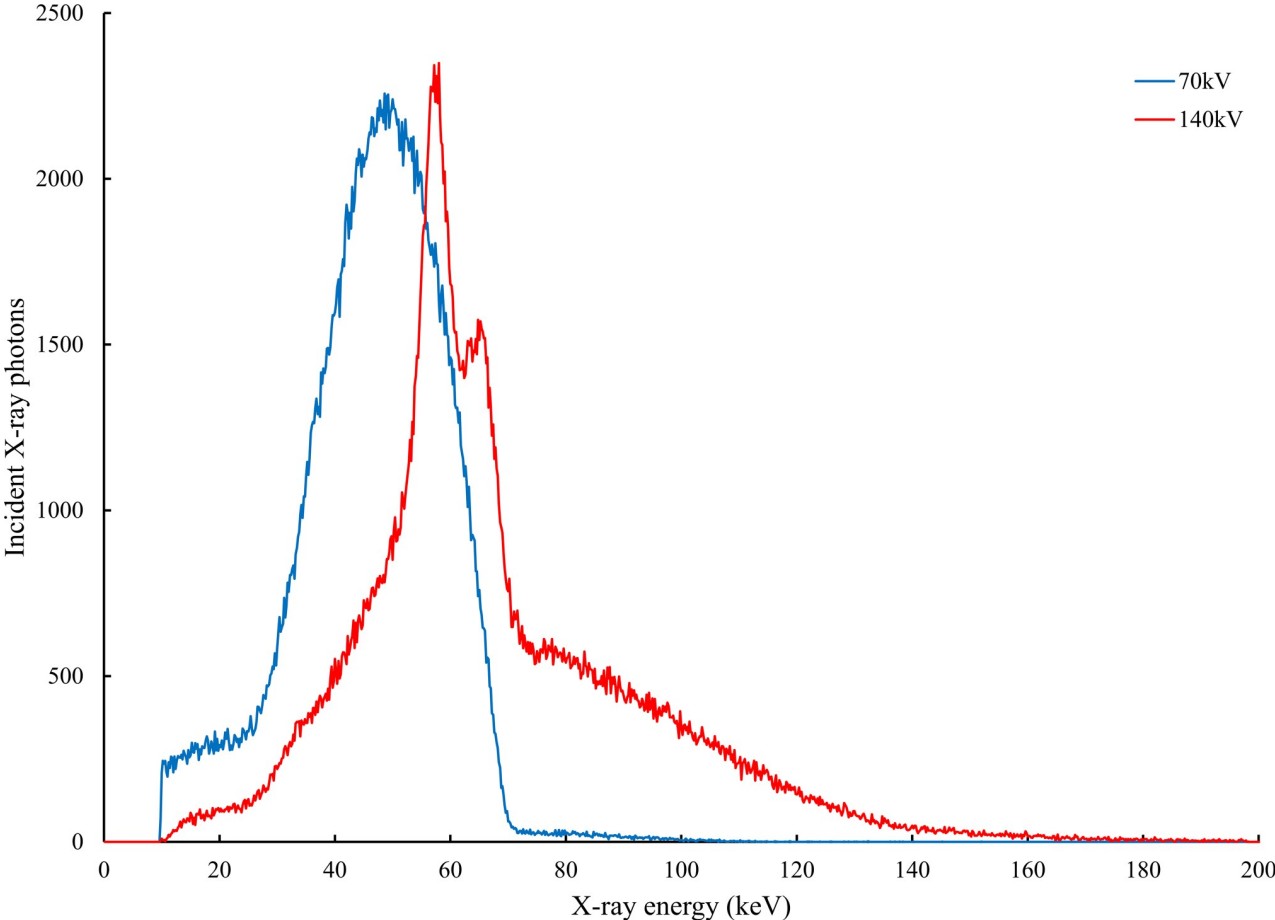

**Fig 3. Spectra of the Sonial Vision Safire II tube at 70 and 140 kV potentials.** The peaks represent the characteristic lines of the tungsten with rhenium and molybdenum anode and the continuous spectrum is a result of Bremsstrahlung. The mean photon energies are 49 and 80 keV, respectively. (Real filter: inherent; aluminum [1.1 mm], additional; aluminum [0.9 mm] and copper [0.1 mm]).

images; $F_t$, $F_w$, $F_f$) were obtained from the inverse of this matrix, as shown in Eq (5):

$$\begin{bmatrix} F_t \\ F_w \\ F_f \end{bmatrix} = \begin{bmatrix} P_{L\_t} & P_{L\_w} & P_{L\_f} \\ P_{H\_t} & P_{H\_w} & P_{H\_f} \\ 1.0 & 1.0 & 1.0 \end{bmatrix}^{-1} \begin{bmatrix} XDTS_{EL} \\ XDTS_{EH} \\ 1.0 \end{bmatrix} \tag{5}$$

$$F_t P_{L\_t} + F_w P_{L\_w} + F_f P_{L\_f} = XDTS_{EL}$$

$$F_t P_{H\_t} + F_w P_{H\_w} + F_f P_{H\_f} = XDTS_{EH}$$

$$F_t + F_w + F_f = 1.0$$

where, two DT projection image sets, each acquired with a different energy ($XDTS_{EL}$ and $XDTS_{EH}$ i.e., 70 and 140 kV).

Inverse of this matrix was used to obtain material fractions. Following decomposition by matrix inversion, the "*inv*" function available in MATLAB (Mathworks; Natick, MA, USA) was employed; this function limits the possible fraction to [0,1] while imposing a sum of 1.

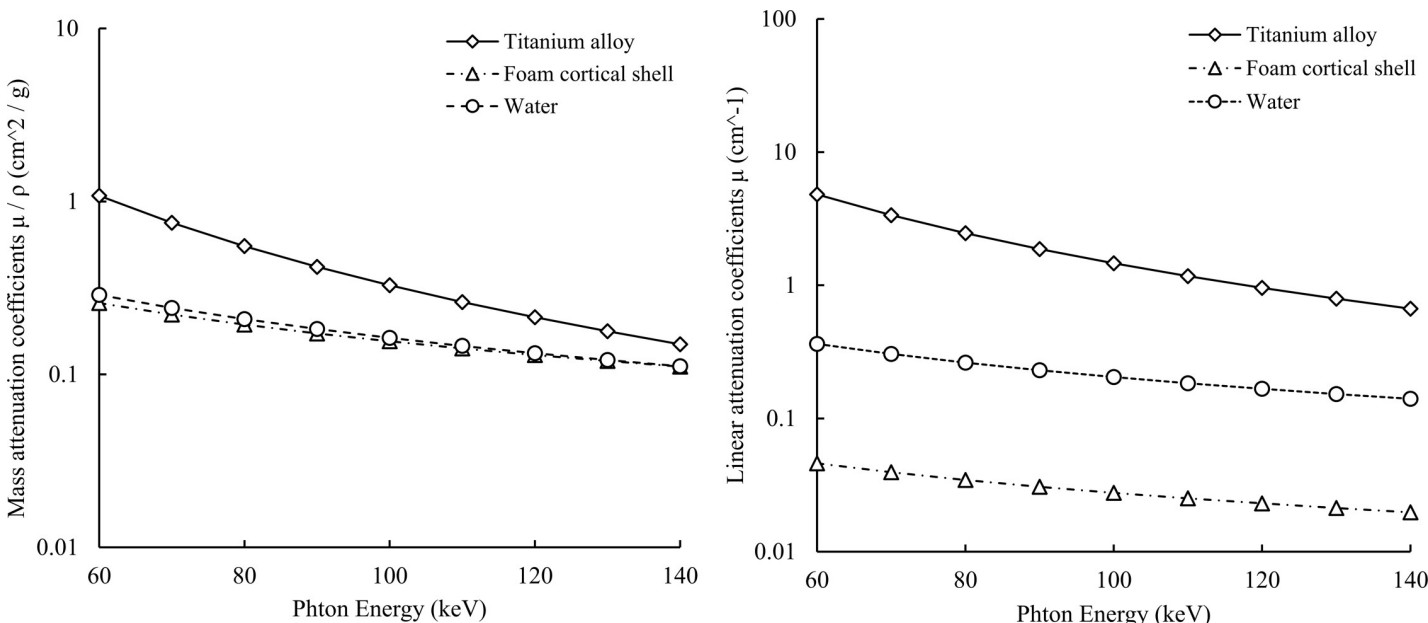

**Fig 4. The linear attenuation and mass attenuation coefficients of a foam cortical shell, titanium alloy, and water with respect to the photons.** Based on the linear attenuation coefficient map, each energy image of virtual monochromatic X-ray processing was created.

Thus, three material fractions arise from the processing pipeline, related to water, foam cortical shell and the titanium alloy [16]. VM processing are obtained using the Eq (4):

$$VM_{-p-img} = F_t * \left(\frac{\mu}{\rho}\right)_t (E) + F_w * \left(\frac{\mu}{\rho}\right)_w (E) + F_f * \left(\frac{\mu}{\rho}\right)_f (E) \qquad (6)$$

where $VM_{-p-img}$ is the VM projection image, and $(\mu/\rho)_t(E)$, $(\mu/\rho)_w(E)$, and $(\mu/\rho)_f(E)$ are each material's corresponding mass attenuation coefficients. The energy of the VM X-ray was selected as 140 keV, which is effective for reducing metal artifacts [16].

## DnCNN-MARHR

Zhang et al. further investigated the construction of a feed-forward DnCNN to include the progresses in learning algorithms, very deep architecture, and methods of regularization for image denoising [19]. Particularly, residual learning and batch normalization were used to fasten the process of training and also to enhance the denoising performance [19]. This network is primarily designed to remove noise from the image (residual learning method). However, it is possible to train the DnCNN architecture to remove artifacts and increase the image resolution. Therefore, by applying this algorithm as an image quality improvement method (for MAR), it is expected that MAR can be effectively achieved. The training workflow of the DnCNN can be realized by using the mini-batch stochastic gradient descent algorithm with momentum (SGDM) method [33]. The SGDM has extensively been employed for the training of CNN models. Even though, the mini-batch SGDM is simple and effective to use, its efficiency of training is mostly compromised by the shift of internal covariates, i.e., alterations in the distributions of internal non-linearity inputs while training [34]. The main algorithm we propose involves processing and correction at the projection data level using the DnCNN. The input image to be corrected was selected to be a low-energy projection image ($P_L$; tube voltage, 70 kV) in consideration of the influences of tube voltage in clinical use and metal artifacts in polychromatic X-ray imaging [16]. In the presence of high-energy, the difference in the linear

attenuation coefficient between the normal tissues becomes narrow, and the contrast tends to decrease. Accordingly, low energy was selected in terms of retention of contrast in normal tissue as well as MAR. The average mean squared error between the required residual projection images and the estimated images from included artifact projection (with noise) image input can be implemented as the loss function $q$ to learn the trainable parameters $\delta$ (SGDM with the weight decay of 0.0001, momentum of 0.9, initial learning rate of 0.1, and hyper-parameters mini-batch size and epochs) in the DnCNN [19]. With regard to $\delta$, mini-batch size and epochs are hyper-parameters that affect MAR [23, 24]. We evaluated the optimal parameter, and we have applied the parameters in the "Evaluation" section below.

$$q(\delta, \varphi) = \frac{1}{2N} \sum_{j=1}^{N} \left\| U(P_{L_j}; \delta) - (P_{L_j} - VM_{-p-img_j}) \right\|_G^2 \qquad (7)$$

where learned a residual mapping $U(P_L)$, $\left\{ (P_{L_j}, VM_{-p-img_j}) \right\}_{j=1}^{N}$ represents $N$ artifact-free training projection image (patch: $N = 32$) pairs and output trained network $\varphi$.

With regard to deep architecture [19], considering a DnCNN with depth $D$, there are three types of layers as follows: (1) convolution + rectifier linear unit (ReLU) [35] (the first layer), 64 filters of size $3 \times 3 \times 1$ are used to generate 64 feature maps and ReLU (max[0, ·]) is then utilized for non-linearity; (2) convolution + batch normalization + ReLU (layers 2~[$D$−1]), 64 filters of size $3 \times 3 \times 64$ are employed and between convolution and ReLU, batch normalization [34] is added; (3) convolution (the last layer), filters of size $3 \times 3 \times 64$ are used to rebuild the output. In this study, network depth was set to 20, scanning step size (horizontal and vertical directions) was set to "1," and the zero padding size was set to "1" in each of the positions (upper, lower, left, and right positions) in the DnCNN [19].

The loss function $q$ in Eq (7) is deployed to learn the residual mapping $U(P_L)$ for residual prediction. The trained network $\varphi$ is activated to estimate the residual projection image.

$$Img_{-Res} = \beta \, \varepsilon \sum_{Z=1}^{Z} (\overline{f_Z} * \varphi_Z(f_Z * P_L)) \qquad (8)$$

where $Img_{-Res}$ is the estimated residual of $VM_{-p-img}$ with respect to $P_L$, $\beta$ is the scanning step size, $\varepsilon$ is the regularization parameter, $f_z$ is the $z$th convolution filter kernel, and $\overline{f_z}$ is the adjoint filter of $f_z$. The influence function $\varphi_z(\cdot)$ can be regarded as pointwise nonlinearity applied to the convolution feature map Eq (8). Eq (8) is a two-layer feed-forward CNN. The CNN architecture further generalizes one-stage trainable nonlinear reaction diffusion (TNRD) [36, 37] from three aspects:

(1) Replacing the influence function with ReLU to ease CNN training;

(2) Increasing the CNN depth to improve the capacity in modeling image characteristics;

(3) incorporating batch normalization to boost the performance.

The connection with one-stage TNRD provides insights into the use of residual learning for CNN-based image restoration [19].

The estimated residual projection image is subtracted from the original projection image to obtain an artifact-reduced projection image.

$$cor_{-img} = P_L - Img_{-Res} \qquad (9)$$

Different reconstruction algorithms have been examined for achieving the metal artifact reduction in arthroplasty images. Among these, a superior performance was obtained using the MLEM method [9]. A two step- MLEM algorithm, that comprises of a forward step (for modeling the DT acquisition process) and a backward step (for updating the reconstructed

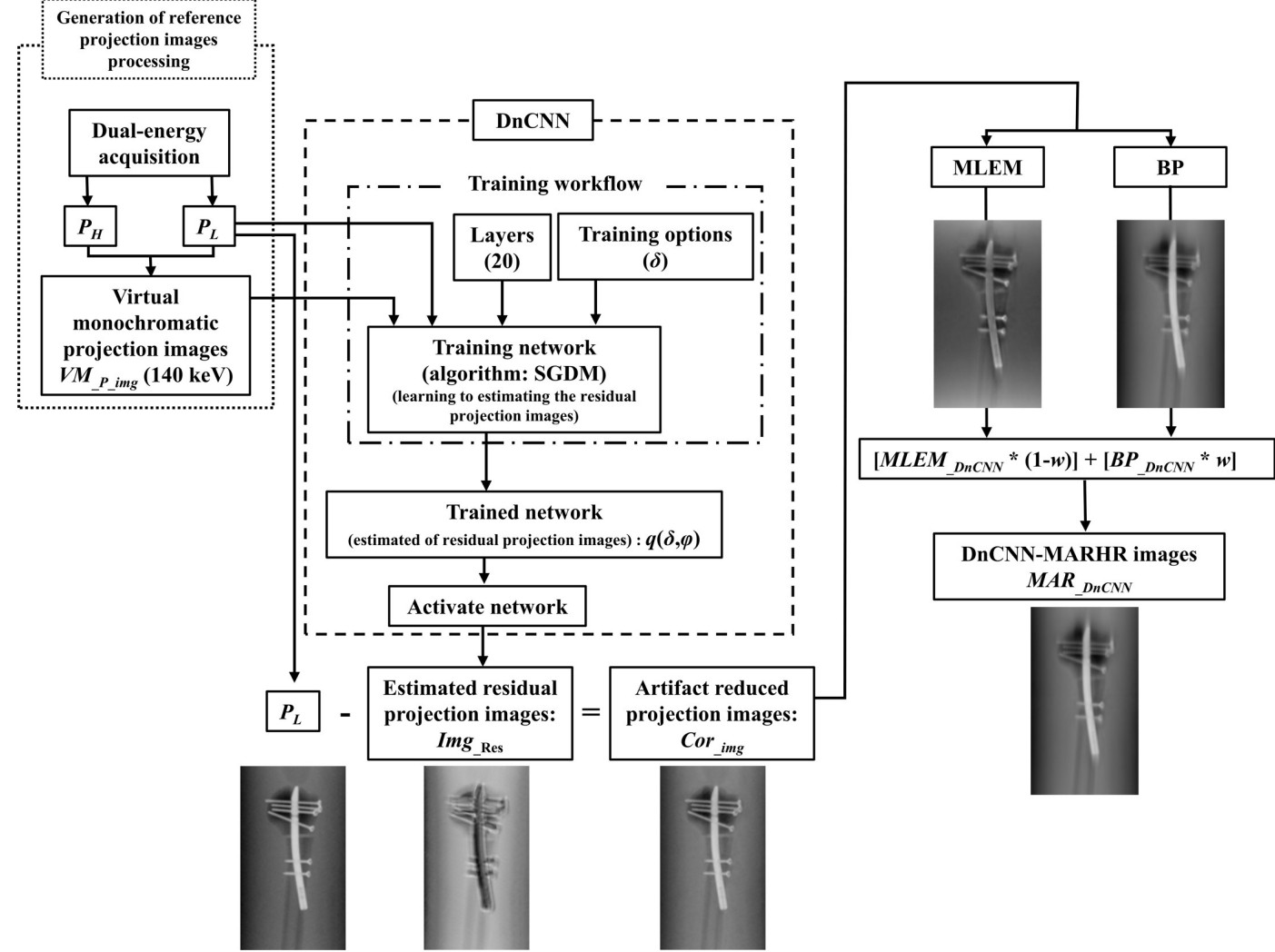

**Fig 5. Flowchart of the denoising convolutional neural network metal artifact reduction hybrid reconstruction (DnCNN-MARHR) algorithm.** The DnCNN-MARHR algorithm was employed to decrease metal artifacts in weighted hybrid reconstructed images (maximum likelihood expectation maximization [MLEM] and back projection). This was achieved using a training network (mini-batch stochastic gradient descent algorithm with momentum) to estimate residual reference images and object images with projection data and subtract the estimated residual images from the object images. Abbreviations: DnCNN = denoising convolutional neural network, SGDM = mini-batch stochastic gradient descent algorithm with momentum, BP = back projection.

object) per iteration has also been proposed [14]. The MLEM algorithm is employed iteratively, so that there is resemblance between the reconstructed volume projections, deduced from an image formation model, and the experimental projections. Thus, we used the MLEM algorithm for reconstructing the projection images with much lower artifacts. Besides, as the noise tends to be more in artifact-curtailed projection data, we suggested the MAR (adaptive filtering) processing to be employed [15] during this reconstruction process. We anticipate a further decrease in both noise and metal artifacts by MAR processing (Fig 5).

The following algorithm for reconstruction and adaptive filtering processing [15] for the real space (post-reconstruction processing) was employed:

$$MLEM_{-DnCNN_{u+1}} \leftarrow MLEM_{-DnCNN_u} \frac{\left[X^T \frac{cor_{-img}}{MLEM_{-DnCNN_u}}\right]}{X^T 1} \qquad (10)$$

$$BP_{-DnCNN} \leftarrow X^T(cor_{-img}) \tag{11}$$

$$MAR_{-DnCNN} = [MLEM_{-DnCNN} * (1 - w)] + [BP_{-DnCNN} * w] \tag{12}$$

The respective parameter definitions used in Eqs (10), (11) and (12) are shown below.

Initialize: $MLEM_{-DnCNN_0}$; all voxels can be initialized to 1.0.

$X^T$ is back projection, $X$ is multiplication by the system matrix, $T$ (superscript) is the matrix response, $u$ is the number of iterations, $MLEM_{-DnCNN}$ is the MLEM [iteration (convergence): 30] image from artifact-reduced projection, $BP_{-DnCNN}$ is the back-projection image from artifact-reduced projection, $MLEM_{-DnCNN}$ is the DnCNN-MARHR image, and $w$ is the weighting coefficient.

## Evaluation

### Optimization parameters for mini-batch size, epochs, and weighting coefficient ($w$)

According to Gomi et al. [9, 15], a weighting coefficient of 0.7 is optimal for effectively processing MAR. Considering these results, the initial weighting coefficient (assumption) was set to 0.7. In the study by Zhang et al. [19], evaluations were performed by setting epochs to 50. In addition, it has been reported that effective MAR can be realized by increasing epochs in terms of MAR [23, 24]. It is considered necessary to increase epochs to realize effective MAR according to these reported results. In this study, the initial epochs (assumption) value was set to 60 considering reports of a value of 50 or more for MAR [19, 23, 24]. The initial value (assumption) was set as follows: epochs, 60 and weighting coefficient ($w$), 0.7, and the mini-batch size optimization and validity of the assumed initial value (epochs and weighting coefficient) setting were evaluated. We set the patch size as 32 × 32, and crop [mini-batch size] × [maximum number of iterations: epoch × number of projection (37)] patches to train the model.

Reconstruction was accomplished using DT system-derived real projection data. MATLAB (Mathworks) was employed for image reconstruction and processing. Artifact index (AI) was deduced for evaluating the effect of MAR on each image in the in-focus plane [38]. Optimization was evaluated using the AI, and the lowest AI value and standard error were selected as the optimum parameters.

### Evaluation of MAR

We calculated the AI to assess the effect of MAR on each in-focus plane image. To compare the difference between DnCNN-MARHR and the conventional algorithm, the difference was evaluated using differential images and mean square error (MSE) in the in-focus plane. We further ascertained the artifact spread function (ASF) to determine the influence of metal artifacts on the features of rebuilt image in the neighboring out-of-plane zone [14]. Texture analysis [39] was used to evaluate the homogeneity of the entire image including metal artifacts. Comparison with the DnCNN-MARHR algorithm was performed by selecting an effective algorithm for MAR (DEMDRA, VM [140 keV], polychromatic MLEM-MAR [70 kV], polychromatic FBP (kernel: Shepp & Logan)-MAR [70 kV], and polychromatic simultaneous algebraic reconstruction technique-TV [SART-TV] [13] MAR [140 kV]) reported by Gomi et al. [16]. The previously reported iteration numbers (DEMDRA, VM, MLEM; 30, SART-TV; 10) were applied to these conventional algorithms for MAR [9]. An iteration number for TV minimization of 100 and length per gradient-descent step of 50 were considered optimal

parameters for SART-TV [9]. DnCNN was evaluated using the application image of optimization parameters.

### Artifact index (AI)

Quantification of the degree of metal artifacts, was done using the AI, which permits low-frequency artifact determination. The AI of identified metal artifacts was as follows:

$$AI_{-n} = sqrt(|Artifact\_ROI\_n - BG\_ROI|) \tag{13}$$

where $n$ = 1, 2,..., 10 defines the formula for *Artifact_ROI_n* and *Artifact_ROI_*1, *Artifact_ROI_*2,..., *Artifact_ROI_*10 that represent the corresponding regions of interest (ROIs) for the relative standard deviations (SDs) of real features (metal artifacts) in the in-focus plane. *BG_ROI* is the relative SD of the background in the in-focus plane [Fig 6 (a)]. For evaluation of the each features (metal artifacts) and background, the ROI was set at 4 × 14 pixels.

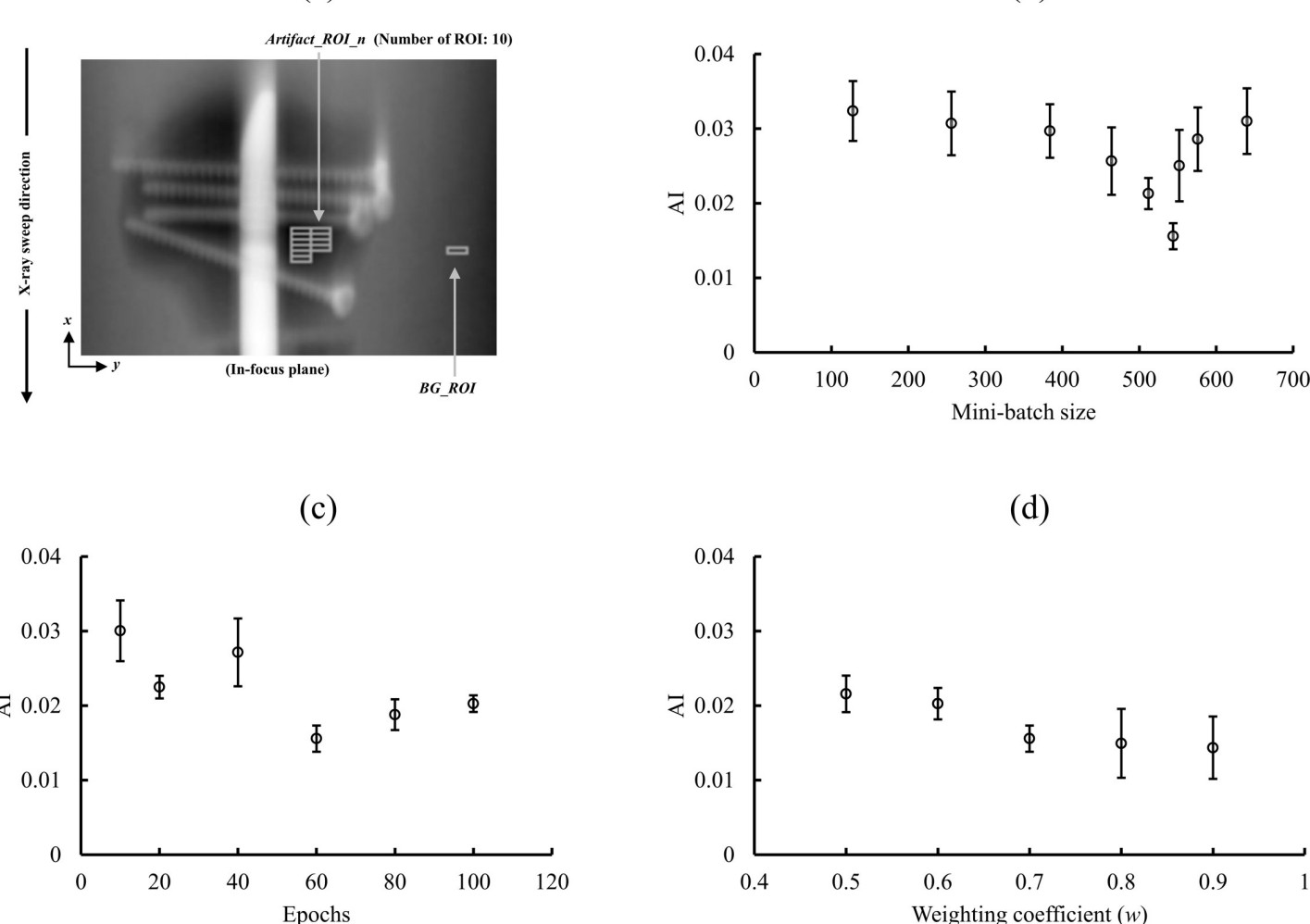

**Fig 6. ROI setting diagram for AI calculation and optimization verification results.** (a) Metal artifacts obtained employing the artifact index (AI) of the selected characteristics. The in-focus plane image displays the background areas and metal artifact of the AI measurements. The AIs resulting from differences in the mini-batch size (b), epochs (c), and weighting coefficient (*w*) in the denoising convolutional neural network metal artifact reduction hybrid reconstruction (DnCNN-MARHR) images are shown (d). The settings of mini-batch size of 544, epochs of 60, and weighting coefficient (*w*) of 0.7 generated the maximum artifact decreasing effect. The error bar represents the standard error.

## Mean square error (MSE)

The MSE of identified in-focus plane image is given as below:

$$MSE = \frac{1}{mn} \sum_{i=0}^{m-1} \sum_{j=0}^{n-1} [K_{ref}(i,j) - V_{obj}(i,j)]^2 \tag{14}$$

where, $K_{ref}(i,j)$ is the $(i,j)$th entry of a DnCNN-MARHR image and $V_{ref}(i,j)$ is the $(i,j)$th entry of an each conventional algorithm image.

## Artifact spread function (ASF)

Wu et al. proposed An ASF metric has been proposed by Wu et al., and this takes into account not only the objects in the focus plane that cause artifacts, but also the effects on these objects, arising from out-of-plane [14]. The resultant measure of ASF reveals the DT's capability to distinguish superimposed features along the direction of tomographic slice. The ASF of artifacts in the image plane is given as below:

$$ASF = \frac{Artifact\_ROI(z) - BG\_ROI(z)}{Artifact\_ROI(z_0) - BG\_ROI(z_0)} \tag{15}$$

where $z_0$ and $z$ represent the positions of the real features (i.e., metal artifacts) in the in-focus and out-of-plane images, respectively; the ROIs for the mean pixel intensities of the features and background in the in-focus-plane image are shown as $Artifact\_ROI$ ($z_0$) and $BG\_ROI$ ($z_0$), respectively; and in the out-of-plane image, the corresponding ROIs are $Artifact\_ROI$ ($z$) and $BG\_ROI$ ($z$). The size of ROI for assessing the background and features (i.e., metal artifacts) was 4 × 14 pixels.

## Texture analysis

The gray-level co-occurrence matrix (GLCM) is a statistical texture inspection method that takes the spatial relationship of pixels into consideration. The GLCM function creates a GLCM by calculating how often a pair of pixels with a specified value and a specified spatial relationship occurs in an image. By extracting statistical information from this matrix, the features of the texture of the image can be obtained. Texture analysis can quantitatively analyze the variation of image intensity in an image [39]. Before computation of texture features, pixel intensities were discretized to 16 gray levels [39]. In addition, pixel values were rescaled between the mean ± SD. The statistical properties of the images derived from the GLCM in this study were evaluated using "inverse difference moment (homogeneity)," "contrast (dissimilarity)," and "correlation," which are defined as follows:

$$Inverse\_Difference\_Moment = \sum_{i,j} \frac{p(i,j)}{1 + |i-j|} \tag{16}$$

$$Contrast = \sum_{i,j} |i-j|^2 p(i,j) \tag{17}$$

$$Correlation = \sum_{i,j} \frac{(i-\mu_i)(j-\mu_j)p(i,j)}{\sigma_i \sigma_j} \tag{18}$$

where $p(i,j)$ is the $(i,j)$th entry of a normalized gray-level spatial dependence matrix, and $\mu_i, \mu_j$ and $\sigma_i$ and $\sigma_j$ are the means and SDs of $p_i$ and $p_j$, respectively.

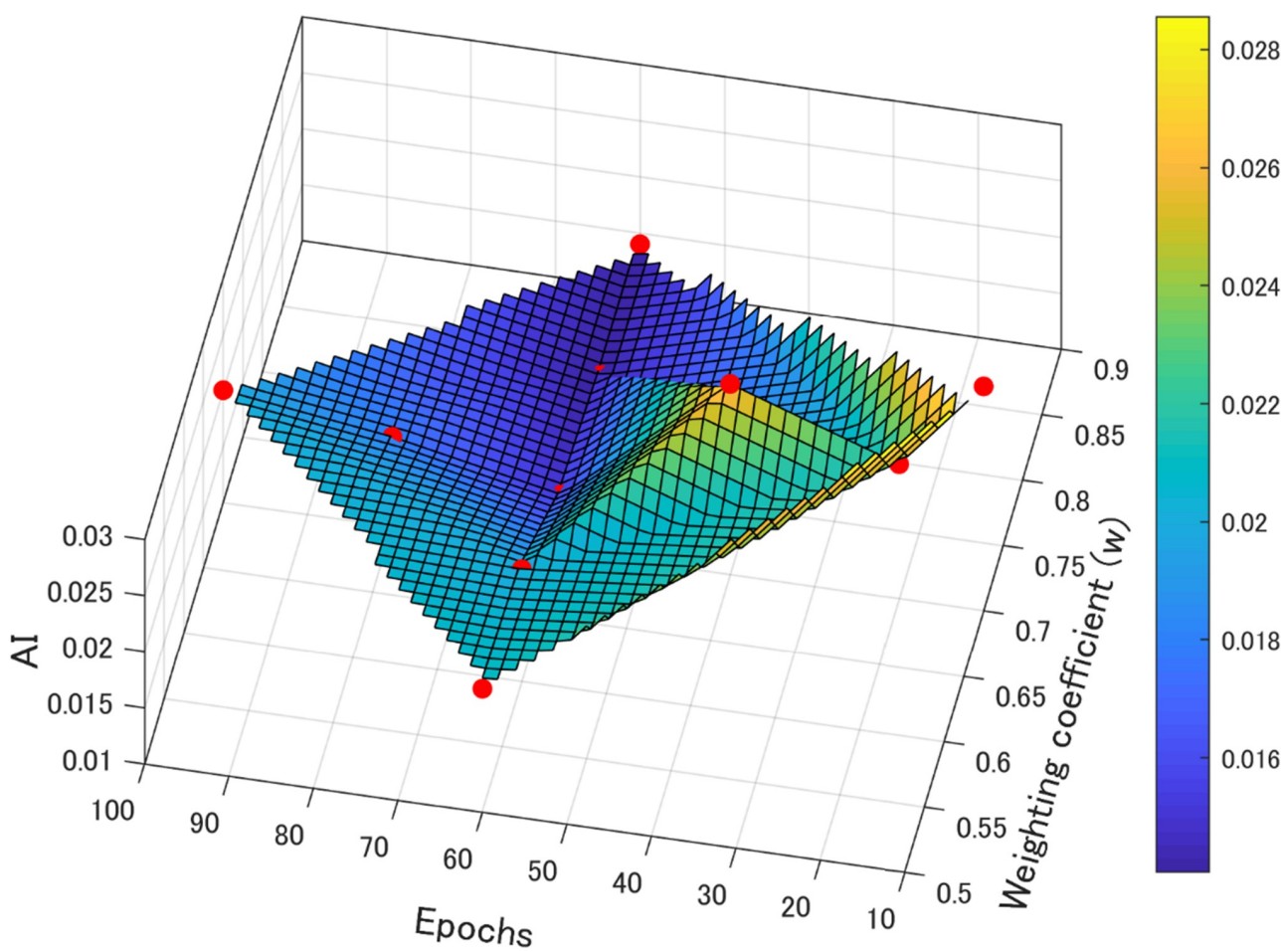

**Fig 7. AI surface plot with epochs and weighting coefficients when mini-batch size at 544.** The surface plot was processed by cubic linear interpolation.

## Results

The initial value (assumption) was set as follows: epochs, 60 and weighting coefficient (*w*), 0.7, and the mini-batch size was changed to 128, 256, 384, 464, 512, 544, 552, 576, and 640. The AI and standard error value of mini-batch size 544 were the lowest [Fig 6(B)]. Next, to verify optimization of the assumed initial value (epochs, 60 and weighting coefficient [*w*], 0.7) with a mini-batch size of 544, the AI and standard error were measured with epochs of 10, 20, 40, 60, 80, and 100, and weighting coefficients of 0.5 to 0.9 [Fig 6(C)]. For epochs of 60 and a weighting coefficient (*w*) of 0.7, the AI value and standard error were the lowest [Fig 6(D)]. Therefore, learning was performed by setting the mini-batch size as 544, epochs as 60 (hyperparameter), and weighting coefficient (*w*) of blended image processing as 0.7 in the DnCNN-MARHR algorithm (Fig 7).

Fig 8 shows the reconstructed images of the prosthetic phantom attained with the DnCNN-MARHR algorithm or each established algorithm for reconstruction with MAR processing. Remarkably, DT images produced employing the DnCNN-MARHR algorithm showed decreased metal artifacts in the X-ray sweep direction (i.e., vertical direction), specifically in the prosthetic phantom's peripheral regions. On the other hand, images produced with the help of FBP-MAR demonstrated more noise and metal artifacts. Comparison of the difference

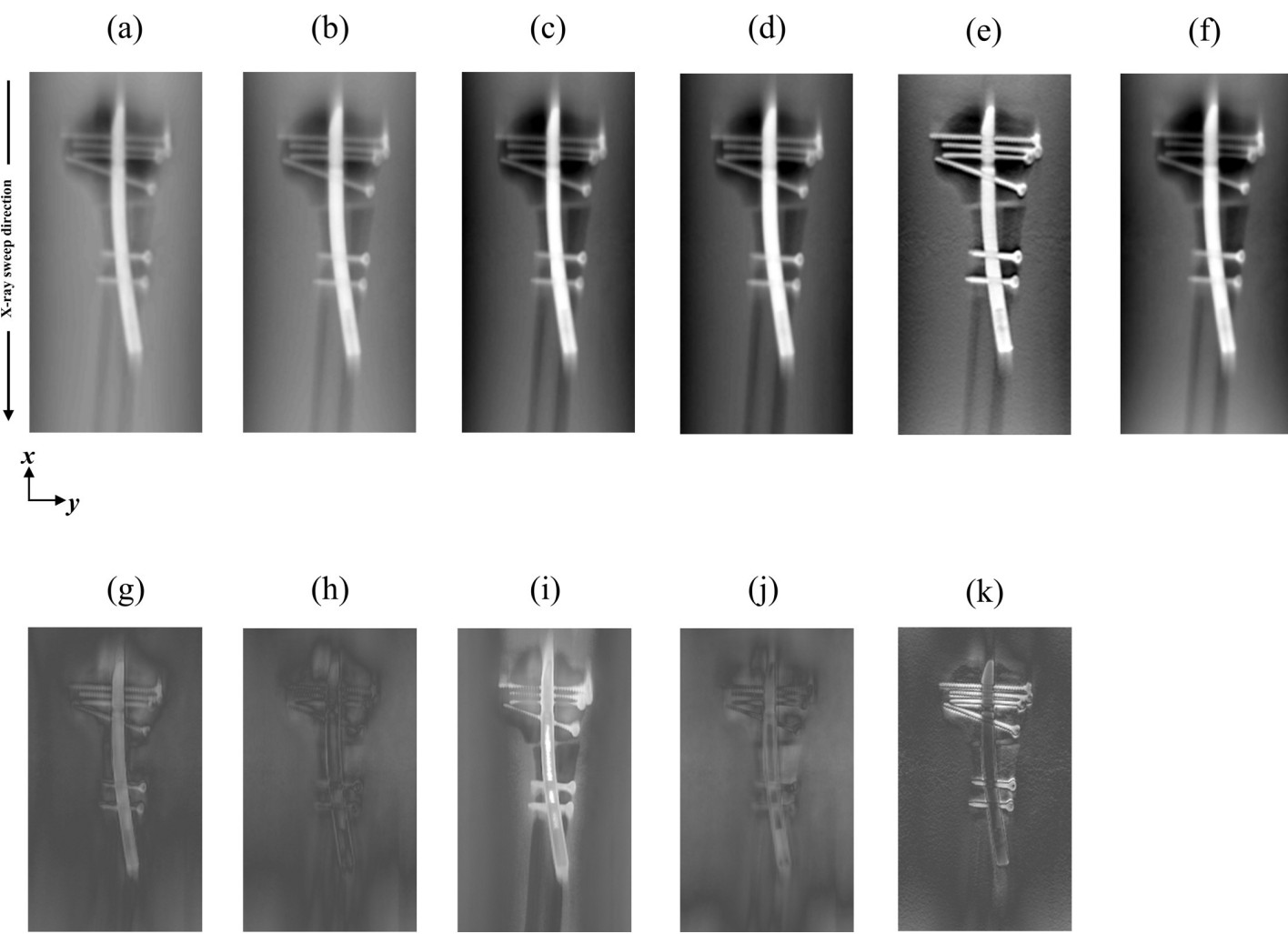

**Fig 8. Comparisons among the denoising convolutional neural network metal artifact reduction hybrid reconstruction (DnCNN-MARHR) algorithm and the traditional reconstruction algorithms with metal artifact reduction (MAR) processing in the in-focus plane.** (DnCNN-MARHR (a), 0.8029–0.9119; DEMDRA (b), 0.9597–0.9914; dual-energy virtual monochromatic [VM]-MAR [140 keV] (c), 0.9487–0.9934; maximum likelihood expectation maximization [MLEM]-MAR [70 kV] (d), 0.9411–0.9983; filtered back projection [FBP, kernel: Shepp & Logan]-MAR [70 kV] (e), 0.5676–0.6117; simultaneous algebraic reconstruction technique-total variation [SART-TV]-MAR [140 kV] (f), 0.7577–0.8184, difference between DnCNN-MARHR and DEMDRA (g), 0–0.1047; difference between DnCNN-MARHR and [VM]-MAR [140 keV] (h), 0–0.1047; difference between DnCNN-MARHR and [MLEM]-MAR [70 kV] (i), 0–0.1047; difference between DnCNN-MARHR and [SART-TV]-MAR [140 kV] (j), 0–0.1047; difference between DnCNN-MARHR and [FBP, kernel: Shepp & Logan]-MAR [70 kV] (k), 0–0.1047) The display variety of the prosthetic phantom is changed to make visual comparison of the contrast and background gray levels. The X-ray source is moved along the image vertically. In the displayed areas, the artifact indices are determined.

between DnCNN-MARHR and the conventional algorithm resulted in the smallest VM (Table 2).

Fig 9 shows the positioning of the ROI in the prosthetic phantom and a graph of the AI results. The DEMDRA gave rise to the smallest values of metal artifact characteristics, irrespective of the status of MAR processing (mean AI ± standard error, 0.01426 ± 0.0022). The difference in AI values between the DnCNN-MARHR algorithm and DEMDRA was small, and the value was lower than that for VM (140 keV) used as a reference image in the training network, confirming the usefulness of MAR (DnCNN-MARHR, 0.01557 ± 0.0017; VM-MAR, 0.01794 ± 0.0025). Metal artifact production was dependent on the reconstruction algorithm type for polychromatic imaging algorithms using MAR processing (MLEM-MAR [70 kV], 0.0195 ± 0.0033; SART-TV-MAR [140 kV], 0.0207 ± 0.0029; FBP-MAR [70 kV], 0.0333 ± 0.0069).

**Table 2. Mean square error (MSE) between each reconstruction algorithm.**

|  | MSE |
| --- | --- |
| **DnCNN-MARHR between DEMDRA** | 3.8048e-05 |
| **DnCNN-MARHR between VM with MAR(140keV)** | 3.5683e-05 |
| **DnCNN-MARHR between MLEM with MAR(70kV)** | 3.7185e-04 |
| **DnCNN-MARHR between SART-TV with MAR(140kV)** | 9.6985e-05 |
| **DnCNN-MARHR between FBP with MAR(70kV)** | 1.0318e-04 |

Results of the comparison of the mean square error (MSE) between the DnCNN-MARHR image and each MAR image. (Comparison image is in-focus plane).

Fig 10 shows a plot of the ASF results and the ROI in the prosthetic phantom. The DnCNN-MARHR algorithm produced the maximum decrease in metal artifacts. However, the FBP-MAR (70 kV) algorithm gave rise to elevated metal artifacts. The DEMDRA, VM-MAR (140 keV), MLEM-MAR (70 kV), and SART-TV-MAR (140 kV) algorithms did not produce any significant alterations in the artifact level.

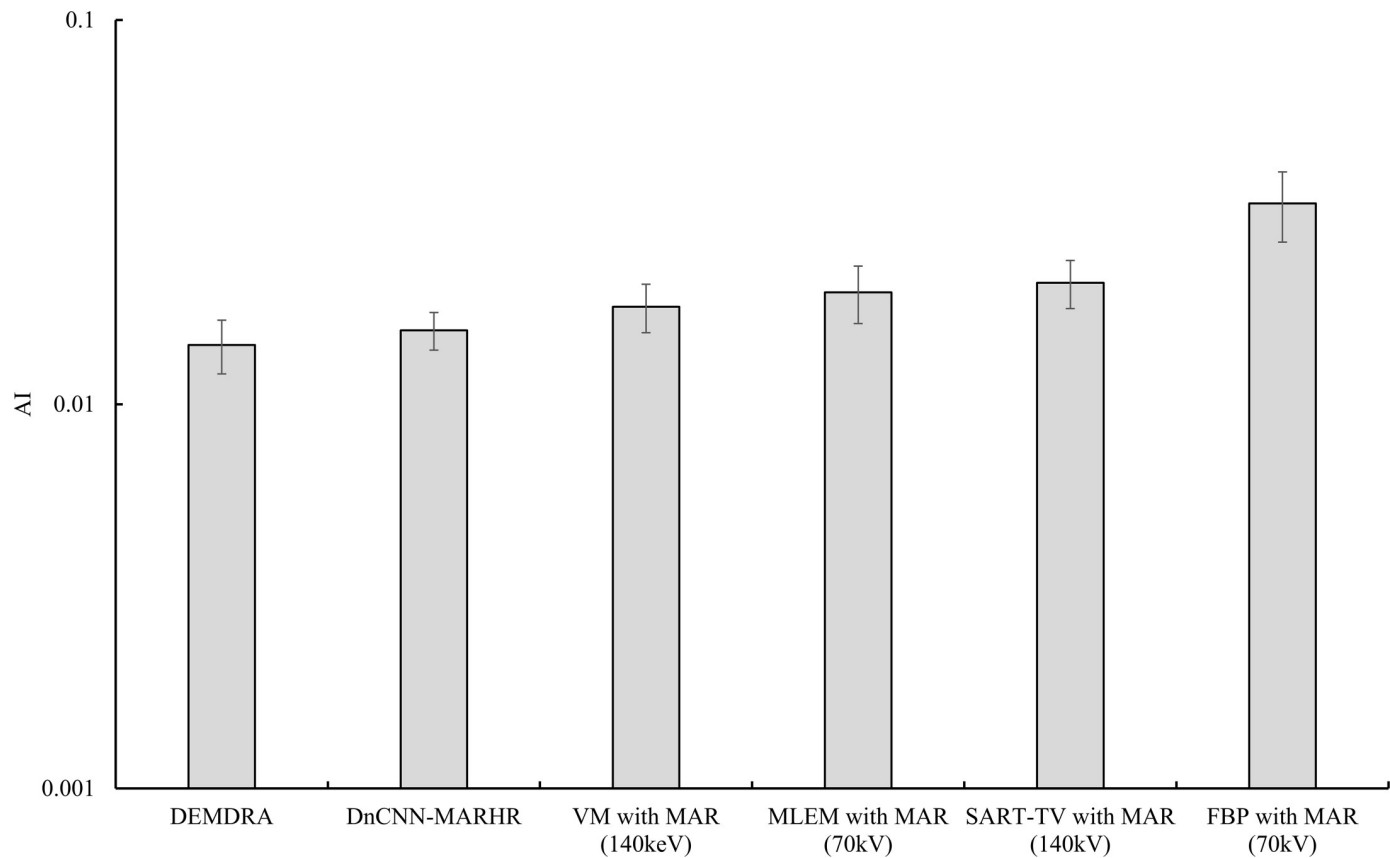

**Fig 9. Comparisons of the artifact indices (AIs) determined for in-focus plane images procured via the denoising convolutional neural network metal artifact reduction hybrid reconstruction (DnCNN-MARHR) algorithm and the traditional reconstruction algorithms with metal artifact reduction (MAR) processing.** Metal artifacts originated from the AIs of 10 selected metal artifact areas (features) and one background area are presented in the in-focus plane. The results are mean ± standard error.

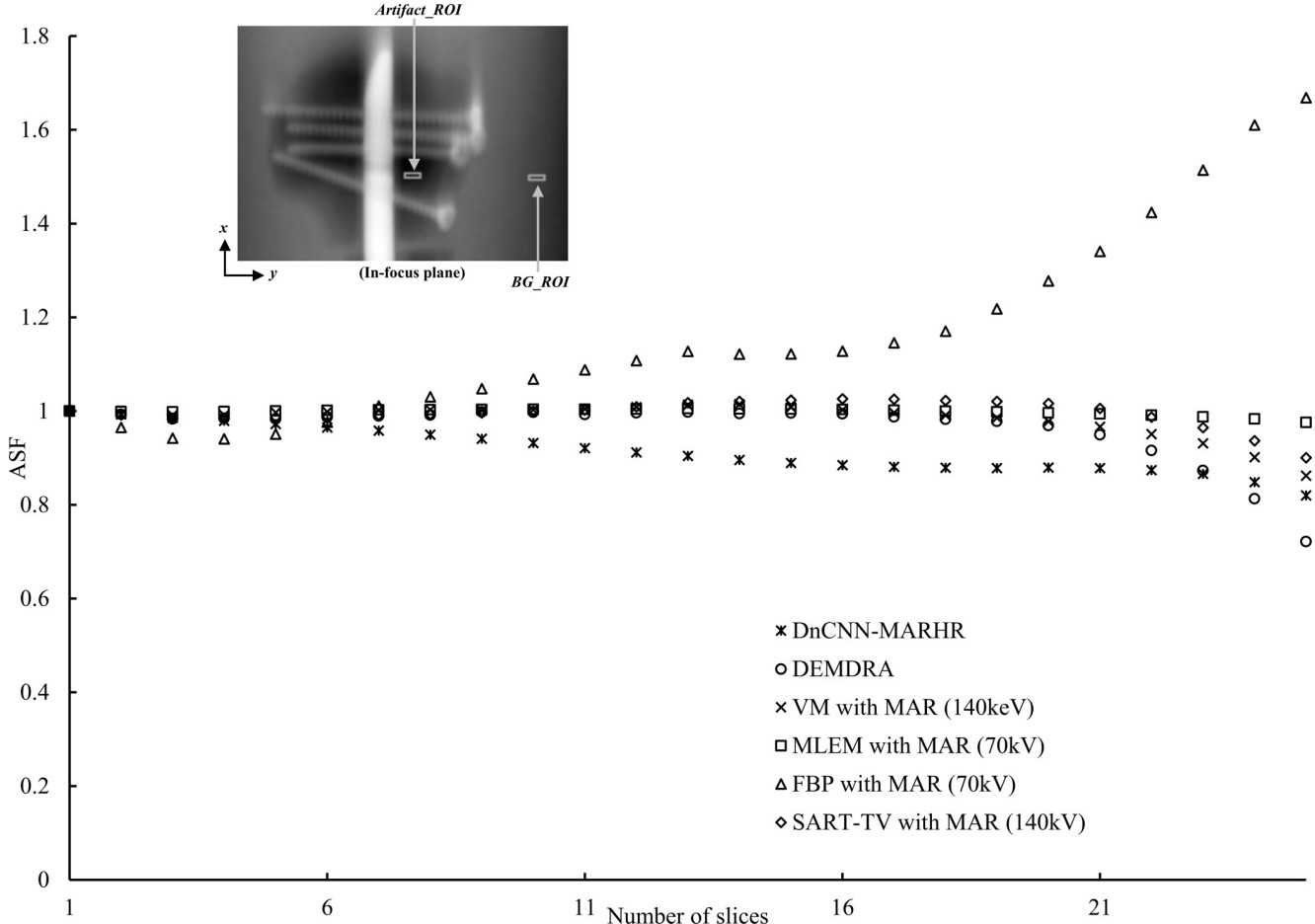

**Fig 10. ROI setting diagram and ASF calculation result for ASF.** (Figure) Metal artifacts derived using the artifact spread function (ASF) of the selected features. The in-focus plane image displays the metal artifact and background areas of the measurements of ASF. (Chart) Plots of the ASF vs. the slice numbers from the in-focus planes of the denoising convolutional neural network metal artifact reduction hybrid reconstruction (DnCNN-MARHR) algorithm and the conventional reconstruction algorithms with metal artifact reduction (MAR) processing.

Fig 11 shows the GLCM calculated using texture analysis with comparison of the feature quantities of "inverse difference moment (homogeneity)" and "contrast (dissimilarity)" and presents the results. The DnCNN-MARHR algorithm showed the best performance regarding the low noise variation image and homogeneous image in the texture analysis (inverse difference moment: DnCNN-MARHR, 0.965; DEMDRA, 0.952; VM-MAR [140 keV], 0.958; MLEM-MAR [70 kV], 0.955; FBP-MAR [70 kV], 0.905; SART-TV-MAR [140 kV], 0.955; and contrast: DnCNN-MARHR, 0.070; DEMDRA, 0.099; VM-MAR [140 keV], 0.085; MLEM-MAR [70 kV], 0.097; FBP-MAR [70 kV], 0.315; SART-TV-MAR [140 kV], 0.104). For the whole image, the measure of the "correlation" between a pixel and the neighboring area was as follows: DnCNN-MARHR, 0.993; DEMDRA, 0.992; VM-MAR (140 keV), 0.993; MLEM-MAR (70 kV), 0.992; FBP-MAR (70 kV), 0.926; SART-TV-MAR (140 kV), 0.990.

Training was implemented by MATLAB on two CPU (Intel(R) Xeon(R) E5-2620 v4, 2.10GHz) and one GPU (NVIDIA Tesla K40c, 12GB) systems. The network required approximately 15 h for training. The reconstruction time was approximately 30 minutes for DnCNN-MARHR (not included in the training network) and DEMDRA approximately 20 minutes for MLEM and SART-TV, and approximately 2 minutes for FBP.

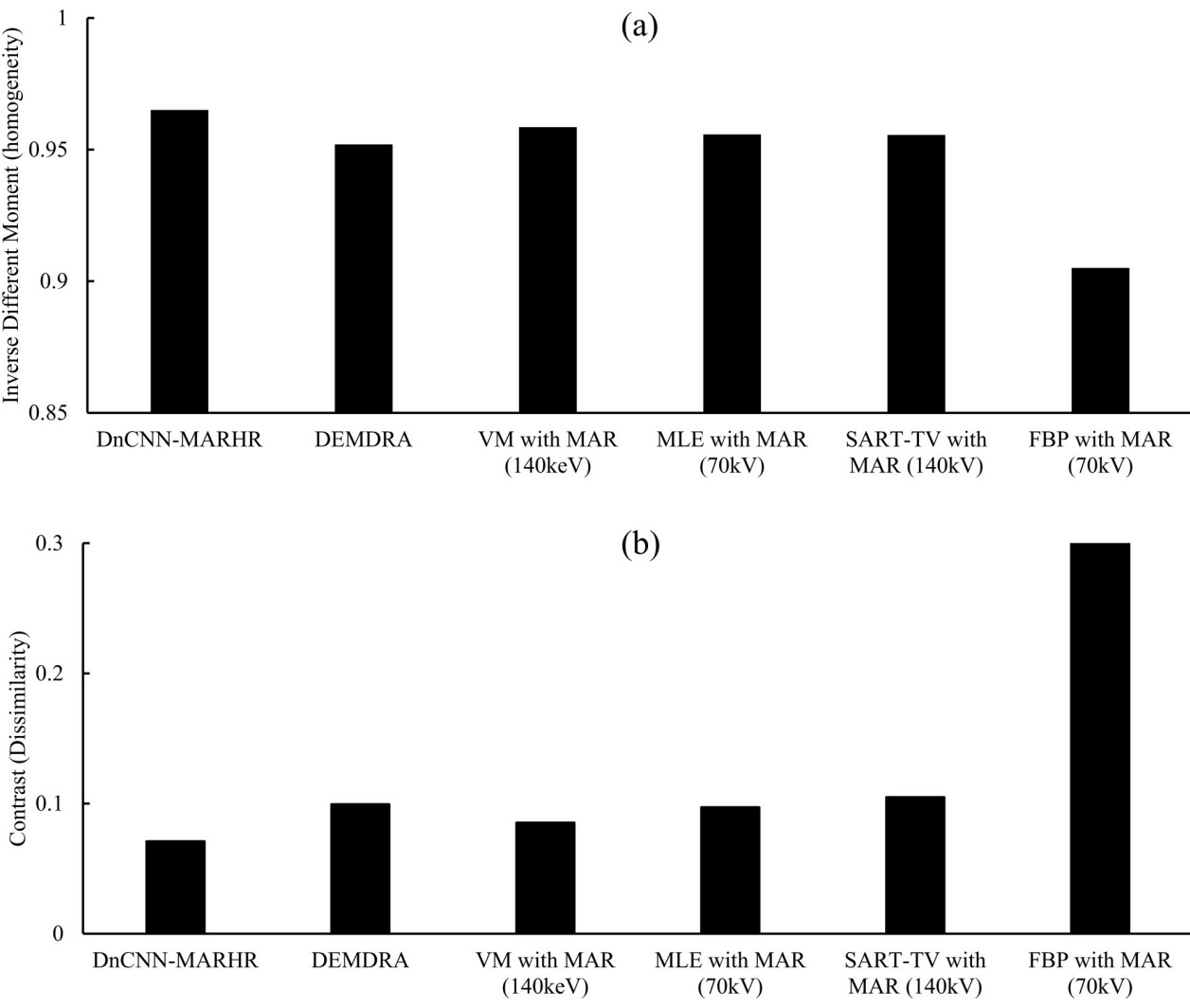

**Fig 11. Texture analysis results.** Comparisons of the inverse different moment (homogeneity) (a) and contrast (dissimilarity) (b) of in-focus plane images obtained via the denoising convolutional neural network metal artifact reduction hybrid reconstruction (DnCNN-MARHR) algorithm and the conventional reconstruction algorithms with metal artifact reduction (MAR) processing.

## Discussion and conclusions

In the present prosthetic phantom study, we compared our DnCNN-MARHR algorithm and different traditional DT reconstruction algorithms without and with MAR processing.

We found that our newly developed DnCNN using the training network algorithm DnCNN-MARHR had adequate overall performance. The DnCNN-MARHR images showed better results that are unaffected by the metal type present in the prosthetic phantom. In addition, this algorithm was efficient in getting rid-off the noise artifacts from images, specifically at higher distances from metal objects. Especially, this algorithm was principally useful for decreasing artifacts related to out-of-plane effects on objects causing artifacts in the focus plane. This algorithm might be a favorable new choice for prostheses imaging, as it yielded 3D visualizations in images with reduced artifacts that were much better than those in images processed employing traditional algorithms. In this DnCNN-MARHR algorithm, the versatility of selecting the imaging parameters, that is dependent on the required final images and

conditions of prosthetic imaging, might be useful to users. Regarding the success or failure of our proposed method, when it is applied in other cases, the key points are the accuracy involved in the VM X-ray process and the method of extraction of the residual image during learning with high accuracy. For example, an effective MAR may not be expected in cases where normal tissues and artificial bones have a complicated arrangement relation.

The proposed DnCNN method should help find a suitable correction map $q:P_L \mapsto Img_{-Res}$ for MAR in such a way that $U(P_L) \approx Img_{-Res}$, where $P_L$ is the attenuation distribution at fixed energy $L$ [19]. The map $q$ should consider $VM_{-p-img}$, as $VM_{-p-img}$ is considered as prior information of DT projection images. Therefore, $q(P_L)$ should be ascertained by not only $P_L$ but also $VM_{-p-img}$. Owing to the highly nonlinear and complicated structure of $VM_{-p-img}$, it could be quite difficult to determine $q$ without using DnCNN methods.

With regard to the DnCNN step, the strength is fusion of helpful facts from various sources to prevent profound artifacts, whereas the limitation is that not all artifacts can be removed, with mild artifacts typically remaining [23]. With regard to adaptive filtering processing in prior image-based MAR methods, moderate artifacts can be removed and a satisfactory prior image can be generated. The success of adaptive filtering processing needs to still be established by imaging, and any noticed artifacts might be due to the absence of the high level normal contributions of artifact-free voxels. Even though, normal contributions are initially made by these voxels, their values decline slightly after removing the largest normal contribution. This means, for each voxel, there will be a rejection of one contribution that is abnormal, while the remaining contributions, along with the largest normal contribution, will be used. Thus, artifact-containing voxels are likely to have elevated values than the surrounding artifact-free voxels. However, when there are major artifacts, the prior image generally suffers from tissue misclassification. By combining the adaptive filtering processing with DnCNN, only with fewer epochs DnCNN training can be stopped, and the procured prior DnCNN is not influenced by tissue misclassification (Fig 11).

The following two factors are important to ascertain exceptional performance of the DnCNN-MARHR algorithm: choosing the right MAR method and training data preparation. Sufficient data for the DnCNN to discriminate artifacts from tissue structures is provided by the appropriately selected MAR method. The preparation of training data ascertains general application of the trained DnCNN by including as many types of metal artifacts as feasible. Thus, it can be considered that the effect of MAR was useful in the longitudinal direction (Fig 10).

TV minimization supposes that a true image is relatively uniform and piece wise. These TV minimization approaches can effectively prevent inadmissible solutions [24], but noise and artifacts appear as deviations or valleys and peaks, which will have comparatively larger TV values since TV is described as the sum of the first-order derivative magnitudes [13], and therefore, their applicability is limited to computed clinical imaging. The DnCNN has the capability to learn nonlinear regression for different sources of artifacts, because it efficiently employs complicated prior knowledge of artifacts and DT images.

In the AI analysis, metal artifacts showed almost an equal MAR effect as the conventional DEMDRA. The SGDM method employed in the DnCNN-MARHR algorithm uses a subset of the training set (mini-batch) to evaluate gradients and update parameters. In general, the DnCNN (or CNN) is usually trained iteratively using multiple image batches, and it is possible to speed up learning without losing accuracy by increasing the mini-batch size. It is considered that reduction of high-frequency metal artifacts cannot be significantly improved owing to the influence of the decomposition/restoration process accompanying the mini-batch and the convolution processing.

Wolterink et al. [22] report that they were useful for noise reduction by using CNN based generative adversarial networks (GAN) in low-dose CT. The results demonstrate that training

with adversarial feedback from a discriminator CNN can produce images that resemble more in appearance to the normal-dose CT than training in the absence of a discriminator CNN [22]. A similar trend was found in our results this time. We think that feedback from avoids smoothing in the image and permits quantification of including metal artifacts objects in DT scans, more accurately.

Our DnCNN-MARHR algorithm has some limitations. First, the algorithm does not take into account metal artifacts that come from photon starvation. The functioning of the learning-based projection data correction method could be perfected by improving the forward model, which precisely represents different realistic artifacts. Second, the suggested learning model is intended for a particular type of phantom implant and it does not work efficiently when the trained network is adapted to projection data correction from totally different scanning geometries. Future studies is required to develop a learning model, which can be used in general cases.

We successfully developed a DnCNN-based algorithm for MAR in DT for arthroplasty. Our DnCNN-MARHR algorithm is particularly useful for reducing artifacts associated with out-of-plane effects on artifact-causing objects in the focus plane, and it is not affected by tissue misclassification.

## Acknowledgments

We wish to thank Mr. Kazuaki Suwa and Yuuki Watanabe at Department of Radiology Dokkyo Medical University Koshigaya Hospital for support on experiment.

## Author Contributions

**Conceptualization:** Tsutomu Gomi.

**Data curation:** Tsutomu Gomi.

**Formal analysis:** Tsutomu Gomi.

**Investigation:** Tsutomu Gomi.

**Methodology:** Tsutomu Gomi.

**Project administration:** Tsutomu Gomi.

**Resources:** Tsutomu Gomi.

**Software:** Rina Sakai, Hidetake Hara.

**Supervision:** Tsutomu Gomi.

**Validation:** Yusuke Watanabe.

**Visualization:** Shinya Mizukami.

**Writing – original draft:** Tsutomu Gomi.

**Writing – review & editing:** Tsutomu Gomi, Hidetake Hara, Yusuke Watanabe.

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
