## [Decision Letter · Decision Letter 0]

26 Jul 2019

PONE-D-19-15441

Development of a novel denoising convolutional neural network-based algorithm for metal artifact reduction in digital tomosynthesis for arthroplasty: A phantom study

PLOS ONE

Dear Prof Gomi,

Thank you for submitting your manuscript to PLOS ONE. After careful consideration, we feel that it has merit but does not fully meet PLOS ONE’s publication criteria as it currently stands. Therefore, we invite you to submit a revised version of the manuscript that addresses the points raised during the review process.

We would appreciate receiving your revised manuscript by Sep 09 2019 11:59PM. To enhance the reproducibility of your results, we recommend that if applicable you deposit your laboratory protocols in protocols.io, where a protocol can be assigned its own identifier (DOI) such that it can be cited independently in the future. For instructions see: http://journals.plos.org/plosone/s/submission-guidelines#loc-laboratory-protocols

We look forward to receiving your revised manuscript.

Kind regards,

Li Zeng

Academic Editor

PLOS ONE

2. Please ensure that you have fully discussed how the present study advances on your previous work in this area. Please ensure that you discuss how your work relates to, and advances upon, the following publication:

"Development of a novel algorithm for metal artifact reduction in digital tomosynthesis using projection-based dual-energy material decomposition for arthroplasty: A phantom study"

https://www.physicamedica.com/article/S1120-1797(18)31136-0/fulltext"

Reviewers' comments:

Reviewer's Responses to Questions

**Comments to the Author**

1. Is the manuscript technically sound, and do the data support the conclusions?

Reviewer #1: Partly

Reviewer #2: Yes

2. Has the statistical analysis been performed appropriately and rigorously? 

Reviewer #1: Yes

Reviewer #2: Yes

3. Have the authors made all data underlying the findings in their manuscript fully available?

Reviewer #1: Yes

Reviewer #2: No

4. Is the manuscript presented in an intelligible fashion and written in standard English?

Reviewer #1: Yes

Reviewer #2: Yes

5. Review Comments to the Author

Reviewer #1: The authors are developing a novel denoising convolutional neural network metal artifact reduction hybrid

reconstruction (DnCNN-MARHR) algorithm for decreasing metal objects in digital tomosynthesis (DT) for

arthroplasty employing projection data. The DnCNN-MARHR algorithm based on a training network

(mini-batch stochastic gradient descent algorithm with momentum) to estimate residual reference and object

images using projection data and subtract the estimated residual images from the object images, involving hybrid

and subjectively reconstructed image usage (back projection and maximum likelihood expectation maximization.

However, the proposed methodology is not new. The paper simply combines several preexisting and widely available techniques, such as, convolutional neural network, residual learning, mini-batch stochastic gradient descent algorithm, back projection maximum likelihood expectation maximization. In this context, what are the methodological and algorithmic contributions of the work to the research community?

#Comments

1. Over all the paper, I'm not sure the purpose of this paper is denoising or metal artifacts reduction.

2. The structure of the article is very confusing. For instance, the equation of the evaluation criteria (artifact index) is in the

Optimization parameters section and the other two evaluation criterias are in the Evaluation section.

3. The methods section is badly written. Many important aspects in the methodology are left unexplained in the article. The authors state to use low-energy projection image (PL) as input image and VM X-ray image as reference image for training workflow (Page 8, Line 159-160). There is no information on how many total data sets and how may data sets were allocated each to training, validation and testing.

4. It is very confusing in how to generate the reference projection images VM X-ray image? What is the meaning of the Ft, Fw and Ff in equation (5), how to determine the value of Ft, Fw and Ff? And why VM X-ray image can be the reference projection images for traning the DnCNN, there's still artifacts in this image in my opinion.

5. In fig 2, I cannot find the difference between the original image (PL) and artifact reduced projection image (Cor_img) from the DnCNN step. I suggest that the authors can directly use the original image (PL) and the equation (11) to obtain the final MAR image and make a comparison between this MAR image and DnCNN-MARHR image, I think the difference in the image quality between this MAR image and DnCNN-MARHR image is small.

Reviewer #2: The authors proposed a novel denoising convolutional neural network metal artifact reduction hybrid reconstruction (DnCNN-MARHR) algorithm to reduce the metal artifacts in digital tomosynthesis. The proposed method is based on a training network to estimate residual reference and the object images using projection. And then subtract the estimated residual images from the object images to get the artifact reduced projection. At last, the hybrid reconstruction algorithm was used to obtain the reconstructed tomosynthesis image with metal artifact reduction. And the authors executed a phantom study to compare the performance with different methods. Overall, the paper includes extensive work and the comparison studies are very clear. However, there are few limitations and the authors are encouraged to address the following concerns:

1. As far as I know, the material decomposition algorithm using dual-energy imaging technique needs a known material phantom to build the relationship between the projection (attenuation) and the material property in the calibration process. For the testing process, the projection of the unknown object with low and high energy was converted to the basis images with known materials in the projection domain. The virtual monochromatic images (VMIs) are generated by using the basis image and the attenuation coefficient according to the linear relation of the attenuation coefficient combination. Do you use the calibration phantom? Please provide more information on the content.

2. The DnCNN-MARHR algorithm is the major development of the study. And the Eq.(7) is the core to calculate the residual projection image. Please add more explanations or references.

3. The specification of the computing environment? How long does it take for training and testing process? Iteration number of different image reconstruction? The differences in the calculation time cost for DnCNN-MARHR algorithm and other methods.

4. What is the image size of the projection? Does it have any image pre-processing on the projections?

5. DEMDRA has the lowest AI value. Why not use DEMDRA image to be the reference image of the training network?

6. In generally, the metal artifacts could be reduced by increasing tube voltage for 3D X-ray imaging. (M.-J.Lee et al., “Overcoming artifacts from metallic orthopedic implants at high-field-strength MR imaging and multi-detector CT,” Radiographics, vol. 27, no. 3, pp. 791–803, 2007.) Why do you choose the projection with lower tube voltage to be the input image for DnCNN-MARHR algorithm?

7. Do you use any additional filters in the low and high tube voltage? Could you provide the energy spectrum of the low and high tube voltage setting?

8. How do you perform the data acquisition of the dual-energy tomosynthesis imaging? Is the imaging with different image parameters in the sequential process?

9. What is the total slice number of the reconstructed tomosynthesis image? What is the start height of the reconstruction from the detector surface? Where is the focal point during tomosynthesis imaging?

10. Line 313-315 Could you provide 2D AI surface plot with epochs and weighting coefficients when mini-batch size is 544?

11. Line 334-335 In Fig.2, the image of MLEM_DnCNN is the same as the polychromatic MLEM-MAR [70 kV]? And the image of BP_DnCNN is also the same as the polychromatic filtered back projection [FBP] MAR [70 kV]? Why do you combine the image of MLEM_DnCNN and BP_DnCNN? Is it empirical knowledge?

12. Could you provide the schematic diagram to illustrate the relationship of the in-focus plane and the out-of-plane in the Z-axis direction?

13. The execute time of DnCNN-MARHR algorithm (not include the training network)?

14. Line 447-448 the imaging parameters case by case? For the object in the different region or the various kind of object (e.g.: materials,…), the proposed algorithm could deal with it?

Minor corrections:

1. Line 137-138 Please normalize the expression of units. –mm μm

2. Line 175-176 Please double check the formula correctness of Eq.(2) and Eq.(3). Does it miss the exponential mark?

3. Line 238 “Equation (6)”

4. Line 245 “Equation (7)”

6. PLOS authors have the option to publish the peer review history of their article (what does this mean?). If published, this will include your full peer review and any attached files.

Reviewer #1: No

Reviewer #2: Yes: Chia-Hao Chang

---

## [Author Response · Author response to Decision Letter 0]

19 Aug 2019

Inserted into the revised manuscript (PDF).

---

## [Decision Letter · Decision Letter 1]

29 Aug 2019

[EXSCINDED]

Development of a denoising convolutional neural network-based algorithm for metal artifact reduction in digital tomosynthesis for arthroplasty: A phantom study

PONE-D-19-15441R1

Dear Dr. Gomi,

We are pleased to inform you that your manuscript has been judged scientifically suitable for publication and will be formally accepted for publication once it complies with all outstanding technical requirements.

With kind regards,

Li Zeng

Academic Editor

PLOS ONE

Additional Editor Comments (optional):

Reviewers' comments:

Reviewer's Responses to Questions

**Comments to the Author**

1. If the authors have adequately addressed your comments raised in a previous round of review and you feel that this manuscript is now acceptable for publication, you may indicate that here to bypass the “Comments to the Author” section, enter your conflict of interest statement in the “Confidential to Editor” section, and submit your "Accept" recommendation.

Reviewer #1: All comments have been addressed

Reviewer #2: All comments have been addressed

2. Is the manuscript technically sound, and do the data support the conclusions?

Reviewer #1: Yes

Reviewer #2: Yes

3. Has the statistical analysis been performed appropriately and rigorously? 

Reviewer #1: Yes

Reviewer #2: Yes

4. Have the authors made all data underlying the findings in their manuscript fully available?

Reviewer #1: Yes

Reviewer #2: Yes

5. Is the manuscript presented in an intelligible fashion and written in standard English?

Reviewer #1: Yes

Reviewer #2: Yes

6. Review Comments to the Author

Reviewer #1: (No Response)

Reviewer #2: Thank you for all your efforts to include new figures and add more explanations in the manuscript. The current version is much clearer and well supported with the further information provided. However, I suggest adding some definition of the material fractions(F_t, F_w, F_f), XDTS_EL, XDTS_EH in Equation (5). That would help readers to understand the physical meanings of this equation.

7. PLOS authors have the option to publish the peer review history of their article (what does this mean?). If published, this will include your full peer review and any attached files.

Reviewer #1: No

Reviewer #2: No

---

## [Editor Report · Acceptance letter]

4 Sep 2019

PONE-D-19-15441R1 

Development of a denoising convolutional neural network-based algorithm for metal artifact reduction in digital tomosynthesis for arthroplasty: A phantom study 

Dear Dr. Gomi:

I am pleased to inform you that your manuscript has been deemed suitable for publication in PLOS ONE. Congratulations! Your manuscript is now with our production department. 

With kind regards,

on behalf of

Professor Li Zeng 

Academic Editor

PLOS ONE